# VIMEX: A MEMORY-CENTERED TASK DESCRIPTION FRAMEWORK FOR VISION-BASED ROBOTICS

## ABSTRACT

Robotics holds the potential to automate applications such as farming, construction, and elderly care; making food, shelter, and dignity easily accessible for everyone. This moonshot goal requires deploying robots in environments that are a priori unknown and typically uninstrumented (e.g., without optitrack, external reward/reset mechanisms, or digital twins), such as agricultural fields, construction sites, or private dwellings. It also requires the same robot to perform numerous different tasks within such environments, with each task defining its own notions of what an object is and what constitutes a desirable way of interacting with it (i.e., affordances). Motivated by these considerations, this paper presents a task-description framework called Vimex (i.e., Visual Memex) that allows a user to efficiently describe vision-based robotics tasks and the associated objects, parts, and affordances without requiring specialized equipment or training a deep neural network. Within this framework, arbitrary object definitions, anywhere on the spectrum between specific instances to general categories, are established using a small number of RGB images captured by a consumer camera, while part definitions are established using scribble annotations over these RGB images. Arbitrary metadata (i.e., any form of task-relevant information) are then attached to these annotations to form records stored in a memory. Given an RGBD image of a scene, these records are retrieved to define probability distributions of part locations and metadata over 3D coordinates using an association process based on nearest-neighbors. Finally, affordance definitions are established as probabilistic inference routines conditioned on such part and metadata distributions. To demonstrate what these abstractions mean and how they can be used to describe tasks to a robot, experiments that focus on vision-based grasping are presented.

## 1 INTRODUCTION

Consider a robot that needs to make a sandwich in a typical kitchen. It first needs to take out a jar of pickles from the refrigerator, and because the exact 3D location of the pickle jar is a priori unknown it should rely on information from onboard sensors (e.g., images) to localize it. Then, it should grasp the jar and move it out of the refrigerator without collisions. To complete these two tasks, there is no need to treat the jar, its lid, and every single pickle inside as separate objects. The robot should then open the jar, so now the body and the lid need to be treated as separate objects and grasped accordingly. Afterwards, the robot should recognize and localize a pickle to grasp it, which requires all pickles to be treated as separate objects. The shape, size, texture, or color of any two pieces of pickle can show considerable variation, which means a 6DOF pose in reference to a common 3D model template may not be the most suitable representation for this grasping task. The robot should then cut the pickle into circular slices to place inside the sandwich, which spawns many new notions of objects with each cut. The Lagrangian state (i.e., 6DOF pose, velocity, and acceleration) of every pickle slice may again not be an appropriate or tractable representation to keep track of. Notice that the only thing the robot has achieved until now was to put pickles in a sandwich, yet the tasks and semantics (i.e., objects, parts, affordances) involved changed many times. Similar considerations arise in most application domains we want to automate (e.g., agriculture, construction, elderly care), highlighting the need for a "task-description framework" (i.e., a common grammar to tell a robot what to do) whose abstractions accommodate the observations and requirements discussed below.

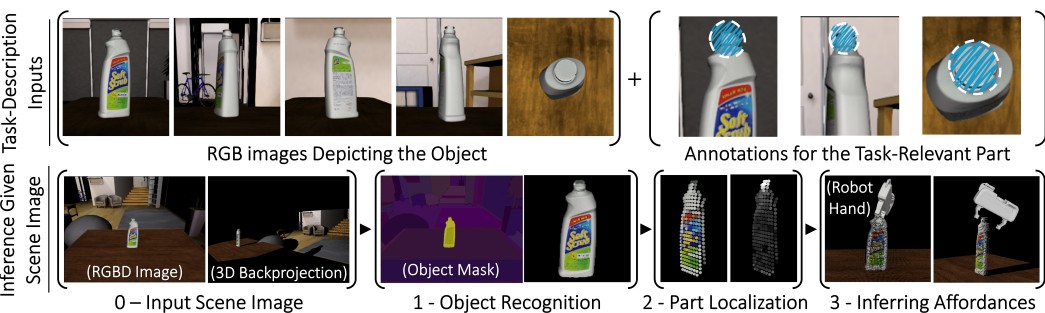

**Figure 1:** This paper aims to provide a framework for describing vision-based tasks to a robot quickly (without training a neural network), easily (without requiring specialized equipment such as optitrack), and efficiently (∼10 RGB images with scribble annotations are sufficient). To describe a task, the user first captures a small number of RGB images that depict an object instance or category. Second, they use a scribble annotation tool to highlight the task-relevant parts of the object and optionally attach any type of metadata to these annotations. Given an RGBD image of a scene, the proposed framework then provides methods to: i) recognize the object, ii) localize its parts and associated metadata in the form of probability distributions over 3D coordinates, iii) infer decision-variables that control robot behavior (e.g., 6DOF grasp poses) conditioned on these distributions.

First, note that every task defines its own notions of objects, parts, and affordances. These definitions do not have to be consistent across different tasks, nor should they remain the same throughout the lifetime of a robot after deployment. If a user wants a cup of coffee, it is important to distinguish the single instance that is their coffee cup from all other cups in their home, and grasp its handle when pouring coffee inside. If the user wants the dishes done, all cups should be considered a single category, and they can be grasped in any way that facilitates scrubbing. Therefore, a task description framework should flexibly accommodate frequent initializations and adjustments of definitions for objects, parts, and affordances. Second, note that the number and variety of tasks that a user may want to utilize the same robot for is a priori unknown, open-ended and ever-changing. Whenever the user comes up with a new task and wants to describe it to a robot (which is quite often), it is impractical for them to collect hundreds of labeled images or kinesthetic demonstrations, engineer a reward/reset mechanism, or build a digital twin of their own kitchen, crop field, or construction site to make adjustments to a neural-network (e.g., train/fine-tune it or modify its architecture to accommodate newly defined tasks). They do not have the training, the tools, the time, or the incentive.

These considerations outline the "task description problem" (Agrawal, 2022), and to the best of our knowledge, state of the art in current robotics literature do not yet provide a satisfactory solution (see Sec.A.1). We therefore propose Vimex, a task description framework that allows describing vision-based robotics tasks (and the objects, parts, affordances involved) with a small set of annotated images without training a neural network or needing specialized equipment (e.g., optitrack).

## 2 RELATED WORK

**Inspirations:** Vimex is inspired by Memex (Bush et al., 1945), a hypothetical precursor to the modern Hypertext (Engelbart, 1963) and WorldWideWeb (Berners-Lee et al., 1994). The two main ideas introduced by Memex were: i) content-based indexing and retrieval of records (rather than address-based), ii) formation of associative trails (i.e., hyperlinks) between records to facilitate cognitive tasks. Both are core principles of Vimex as well. The main observation that motivates this paper (i.e., every task defines its own objects, parts, and affordances) draws from classical texts in vision science (Palmer, 1999) and cognitive science (Cohen & Lefebvre, 2005), which themselves refer to seminal works around categorization (Bruner et al., 1977), the exemplar (Nosofsky, 1986; Kruschke, 1992) and prototype (Rosch et al., 1978; 1976; Tversky & Hemenway, 1984) theories, and the theory of affordances (Gibson, 1977).

**Similar Frameworks:** Most ideas and design choices employed in Vimex are reiterations of concepts that have continually been revisited in computer vision and robotics. Early work by Lowe (1999; 2001) describes a framework that represents each object as a set of its images, and each image as a set of descriptors placed at salient points. This process is then used to build a database of

exemplar objects whose descriptors are matched to the descriptors of test images to recognize objects within. These ideas are central to Vimex as well. Another important idea from the same body of work is the "probability of non-accidental cooccurance" (Lowe, 1985)[1], which provides a qualitative explanation about why descriptors obtained from the foundation model of Oquab et al. (2023) are particularly suitable for establishing spatial correspondences to exemplar images. Subsequent work by Belongie et al. (2002) builds upon the same concepts to create another exemplar-based object recognition system. Of particular importance are the proposed shape context distance, image appearance distance, and bending energy metrics, as Vimex implements functionally analogous terms for nearest-neighbor retrieval. Another major influence, also named Visual Memex by its authors, is presented by Malisiewicz (2011), from which Vimex borrows two main ideas: i) using nearest-neighbor classifiers with a small positive-set and a disproportionally large negative-set to represent objects, and ii) going beyond object recognition to retrieve and transfer other sources of information. Finally, Manuelli et al. (2019) proposes defining affordances in reference to a set of task-relevant 3D keypoints. Inspired by this, Vimex defines affordances in reference to continuous distributions over 3D coordinates instead of discrete 3D keypoints; and instead of training a different keypoint regression network for every task, it prompts (Gu et al., 2023) a foundation model (Oquab et al., 2023) for spatial descriptors. For further elaboration on related work, please see Sec.A.1.

## 3 PRELIMINARIES

**Global and Spatial Descriptors:** In Vimex, the global descriptor $\mathbf{z}_{\mathrm{glb}}$ and spatial descriptors $\{\mathbf{z}_i\}$ of an input image are obtained from the CLS and spatial tokens of the final attention block of DINOv2 (Oquab et al., 2023). The spatial descriptor index $i$ runs over the 2D grid of patches used as input to the ViT-B backbone (Dosovitskiy et al., 2020). The resolution of this spatial descriptor grid can be set arbitrarily by adjusting the kernel-size and stride of the first convolution layer that performs the linear embedding of spatial patches (the positional embeddings of the resulting tokens should also be interpolated accordingly). The important thing about these descriptors is that the inner-product $\mathbf{z}_1^T \mathbf{z}_2$ implements a semantically meaningful nearest-neighbors similarity metric for image retrieval and patch matching. For a qualitative explanation, together with connections to the concept of "probability of non-accidental cooccurance" (Lowe, 1985), please see Sec.A.2.1.

**Class-Agnostic Segmentation:** Object recognition in Vimex requires an object discovery (Tuytelaars et al., 2010; Rubinstein et al., 2013; Burgess et al., 2019) method that can create a set of segmentation mask proposals $\{m_i\}$ in a class-agnostic manner (i.e., binary masks without any semantic labels) given an image. The Segment Anything foundation model (SAM) (Kirillov et al., 2023) is used off-the-shelf for this purpose. We assume that if a task-relevant object is present in an image, its associated segmentation mask will be present in $\{m_i\}$. Given SAM's internet-scale training, we empirically observed this assumption to be valid.

**Appearance and Optimal Transport Distances:** Let $I_1, I_2$ denote two images with corresponding global and spatial descriptors $(\mathbf{z}_{\mathrm{glb},1}, \{\mathbf{z}_{i,1}\})$, $(\mathbf{z}_{\mathrm{glb},2}, \{\mathbf{z}_{j,2}\})$, with $|\{\mathbf{z}_{i,1}\}| = N_1$ and $|\{\mathbf{z}_{i,2}\}| = N_2$. The nearest-neighbor distance used in Vimex is a composition of two terms. The first term, called the appearance distance, is computed as $d_{\mathrm{ac}}(I_1, I_2) = -\max(0, \mathbf{z}_{\mathrm{glb},1}^T \mathbf{z}_{\mathrm{glb},2})$, and serves an analogous function as the appearance cost of Belongie et al. (2002). The second term is the optimal transport distance $d_{\mathrm{ot}}(I_1, I_2)$ between the spatial descriptors of the two images (Peyré et al., 2019). It is computed between two uniform distributions with $N_1$ and $N_2$ bins, using the $N_1 \times N_2$ pairwise cost matrix $\mathbf{C}_{ij} = -\mathbf{z}_{i,1}^T \mathbf{z}_{j,2}$, and with the Sinkhorn-Knopp (SK) algorithm (Cuturi, 2013). This is practically identical to the shape context matching cost of Belongie et al. (2002) (instead of solving the exact assignment problem with the Hungarian algorithm, Kantorovich's relaxation is solved with the SK algorithm). The two distances are combined as $d(I_1, I_2) = -d_{\mathrm{ac}}(I_1, I_2) \, d_{\mathrm{ot}}(I_1, I_2)$.

## 4 VISUAL MEMEX: USER INPUTS FOR TASK-DESCRIPTION

**User Inputs for Object Recognition:** Consider two example robotic grasping scenarios: i) a robot that should distinguish a user's personal cup from all other cups and grasp its handle (requiring instance-level specificity), and ii) a robot that should pick strawberries in a field by grasping their stems (requiring category-level generalization). For the cup example, the user captures ˜5 images of

---

[1]This is named "probability of accidental occurance" in (Lowe, 1985).

their personal cup, and for the strawberry picking example, the user goes out into the field and captures ~50 images of different strawberries, using a consumer RGB camera for both cases. The central object in each image is automatically segmented (Kirillov et al., 2023) to remove the background. The images are then cropped to a tight square around the object and resized to a standard resolution (e.g., $224 \times 224$) to form the set of positive exemplars $\mathbb{D}_{\text{pos}} = \{I_i\}$. In general, the number of images $|\mathbb{D}_{\text{pos}}|$ necessary depends on the intrinsic visual variation between different viewpoints of an instance (Koenderink & Van Doorn, 1979), and between different instances of a category (Belongie et al., 2002). Similar to (Malisiewicz, 2011), a much larger negative exemplar set (e.g., $\sim 10^6$ images) is then created automatically as follows. First, images of a common large dataset such as Deng et al. (2009); Li et al. (2020) are processed the same way as $\mathbb{D}_{\text{pos}}$ (i.e., background removal, cropping, and resizing) to form $\mathbb{D}_{\text{neg}} = \{I_i\}$. Then the median $d(I_i, I_j)$ between all pairs $(I_i, I_j)$ in $\mathbb{D}_{\text{pos}}$ are computed. All images in $\mathbb{D}_{\text{neg}}$ whose nearest-neighbor distance to $\mathbb{D}_{\text{pos}}$ is smaller than this are marked. The user then decides whether the marked images should stay in $\mathbb{D}_{\text{neg}}$, moved to $\mathbb{D}_{\text{pos}}$, or removed entirely. For a final and optional step, the user manually captures and adds any other negative exemplar images they want to $\mathbb{D}_{\text{neg}}$ (e.g., for the cup example, images of all cups other than the user's personal cup can be placed in $\mathbb{D}_{\text{neg}}$ to ensure instance-level specificity). In practice, only the descriptors rather than the full images need to be stored (e.g., ~20 MB for Li et al. (2020)).

**User Inputs for Part Localization** The user annotates the images in $\mathbb{D}_{\text{pos}}$ with a scribble annotation tool to mark task-relevant parts (e.g., the handle of the cup, or the strawberry stems). For the strawberry case, not all images need to be annotated (around ~5 is sufficient). The set of spatial descriptors from all annotated regions for a task-relevant part is denoted as $\mathbb{A} = \{\mathbf{z}_j\}$. Afterwards, two crucial statistics are computed over $\mathbb{D}_{\text{neg}}$. The first one is $c_{\text{mean}}$, the average $\mathbf{z}_1^T \mathbf{z}_2$ between two randomly sampled spatial descriptors from two randomly sampled images in $\mathbb{D}_{\text{neg}}$. The second one is $c_{\mathbb{A}}$, the average of $\mathbb{E}_{\mathbf{z}_j \sim \mathbb{A}}[\mathbf{z}_j^T \mathbf{z}]$ where $\mathbf{z}$ is a randomly sampled spatial descriptor from a randomly sampled image in $\mathbb{D}_{\text{neg}}$. The annotation $(\mathbb{A}, c_{\mathbb{A}})$ and the scalar $c_{\text{mean}}$ are stored for later use in Sec.5.2.

**User Inputs for Metadata Retrieval:** Vimex defines a memory as a set $\mathcal{M} = \{(\mathbf{z}_i, \mathbb{R}_i)\}$ of tuples, where $\mathbf{z}_i$ denotes a spatial descriptor and $\mathbb{R}_i$ contains the associated metadata (i.e., any set of task-relevant information that will later be retrieved and utilized). Considered together as a single tuple, $(\mathbf{z}_i, \mathbb{R}_i)$ is referred to as a record. The exact nature of the metadata $\mathbb{R}_i$ and the associated process of obtaining it naturally depends on the specific task and its affordances. For the grasping experiments in Sec.7, $\mathbb{R}_i = \{d_i\}$ where $d_i$ are the "antipodal distances" (i.e., computed by starting from a point $v_i$ on the object and following the surface normal $n_i$ inwards until the first intersection with the surface on the other side). For more elaboration on measuring $\{d_i\}$ from RGB images, see Sec.A.3.1.

## 5 VISUAL MEMEX: INFERENCE

### 5.1 OBJECT RECOGNITION

The steps involved in object recognition are as follows. First, mask proposals $\{m_i\}$ are obtained using SAM. For each mask proposal $m_i$, the background is removed and a tight square around the mask is cropped and resized to the standard resolution, resulting in an image $I_i$. As in Lowe (1985); Belongie et al. (2002); Malisiewicz (2011), $\{I_i\}$ are then classified using a binary nearest-neighbor classifier based on the metric $d(I_1, I_2)$, and $(\mathbb{D}_{\text{pos}}, \mathbb{D}_{\text{neg}})$ defined before. Because there is a large difference between the sizes of $(\mathbb{D}_{\text{pos}}, \mathbb{D}_{\text{neg}})$, it is crucial to balance their contributions to the nearest-neighbor classification rule (Malisiewicz et al., 2011). We therefore implement an "any top-K" rule: the classification label is positive if and only if there exists any positive exemplar within the top-K nearest neighbors, where K is set through cross-validation within $(\mathbb{D}_{\text{pos}}, \mathbb{D}_{\text{neg}})$. There are multiple benefits of this approach. First, the user only needs a small number of RGB images to build the classifier. Second, detections are always paired with a nearest-neighbor, which provides a good degree of interpretability. As a result, the classifier can easily be modified and calibrated in an active-learning manner (Settles, 2009). For example, to immediately fix a false-negative, all that is necessary is to take a few images of the problematic object and place them in the positive set.

### 5.2 PART LOCALIZATION

Given an RGBD image of a scene $I_{\text{scn}}$, object recognition provides a segmentation mask $m_{\text{obj}}$. Then, pixels inside $m_{\text{obj}}$ are backprojected into 3D and the tight axis-aligned rectangle volume $V_{\text{obj}}$

surrounding these points is computed. We define a function $f_{\mathrm{obj}}$ that assigns a descriptor to every coordinate $\mathbf{x} \in V_{\mathrm{obj}}$ as follows:

- $f_{\mathrm{obj}}(\mathbf{x}) = \mathbf{z}$ if $\mathbf{x}$ lies on the object surface inside $m_{\mathrm{obj}}$, where $\mathbf{z}$ is the spatial descriptor from DINOv2 computed on the pixel projection of $\mathbf{x}$.

- $f_{\mathrm{obj}}(\mathbf{x}) = \mathbf{z}_{\mathrm{occ}}$ if $\mathbf{x}$ belongs to occluded space inside $m_{\mathrm{obj}}$, where $\mathbf{z}_{\mathrm{occ}}$ is a unique identifier.

- $f_{\mathrm{obj}}(\mathbf{x}) = \mathbf{z}_{\mathrm{null}}$ if $\mathbf{x}$ belongs outside $m_{\mathrm{obj}}$ or any empty space in $I_{\mathrm{scn}}$, where $\mathbf{z}_{\mathrm{null}}$ is another unique identifier.

As usual, whether $\mathbf{x}$ is occluded, in empty space, or on the object surface is determined by projecting it onto the RGBD image $I_{\mathrm{scn}}$ and comparing the depth values. Assume that the user provides annotations $\{(\mathbb{A}_n, c_{\mathbb{A},n})\}_{n=1}^{N}$ for $N$ different parts as described in Sec.4. We then define:

$$c_n^*(\mathbf{x}) = \begin{cases} 0 & \text{if } f_{\mathrm{obj}}(\mathbf{x}) = \mathbf{z}_{\mathrm{null}}. \\ c_{\mathbb{A},n} & \text{if } f_{\mathrm{obj}}(\mathbf{x}) = \mathbf{z}_{\mathrm{occ}}, \\ \mathbb{E}_{\mathbf{z} \sim \mathbb{A}_n}[\mathbf{z}^T f_{\mathrm{obj}}(\mathbf{x})] & \text{otherwise.} \end{cases}$$

Let us use the notation $\mathbb{A}(\mathbf{x}) = \mathbb{A}_n$ to denote the probabilistic event that $\mathbf{x} \in V_{\mathrm{obj}}$ belongs to part $\mathbb{A}_n$. Using $c_n^*(\mathbf{x})$, we define the corresponding probability measure as:

$$p[\mathbb{A}(\mathbf{x}) = \mathbb{A}_n] = \frac{e^{c_n^*(\mathbf{x})}}{e^{c_{\mathrm{mean}}} + \sum_{n=1}^{N} e^{c_n^*(\mathbf{x})}}.$$

For visible points on the object surface $m_{\mathrm{obj}}$, this soft-max distribution is an increasing function of the nearest-neighbor descriptor similarity between $f_{\mathrm{obj}}(\mathbf{x})$ and the set $\mathbb{A}_n$. For occluded points, it defaults to the large-scale dataset statistics $c_{\mathbb{A},n}$ computed over $\mathbb{D}_{\mathrm{neg}}$, and for empty points it is zero.

## 5.3 METADATA RETRIEVAL

Given the memory $\mathcal{M} = (\mathbf{z}_i, \mathbb{R}_i)$, let us define the notation $\mathbb{R}(\mathbf{x}) = \mathbb{R}_i$ to denote the probabilistic event that metadata $\mathbb{R}_i$ is associated with $\mathbf{x} \in V_{\mathrm{obj}}$. Using descriptors $f_{\mathrm{obj}}$, we define:

$$p[\mathbb{R}(\mathbf{x}) = \mathbb{R}_i] = \begin{cases} 0 & \text{if } f_{\mathrm{obj}}(\mathbf{x}) = \mathbf{z}_{\mathrm{null}} \text{ or } \mathbf{z}_{\mathrm{occ}}, \\ \frac{e^{\mathbf{z}_i^T f_{\mathrm{obj}}(\mathbf{x})}}{\sum_i e^{\mathbf{z}_i^T f_{\mathrm{obj}}(\mathbf{x})}} & \text{otherwise.} \end{cases}$$

For visible points on the object surface $m_{\mathrm{obj}}$, this soft-max distribution is an increasing function of the nearest-neighbor descriptor similarity between $f_{\mathrm{obj}}(\mathbf{x})$ and $\mathbf{z}_i$. No metadata is associated with occluded points or empty points so the corresponding probability density is zero.

## 5.4 INFERRING AFFORDANCES

Suppose we are provided with an arbitrary set of "visual decision variables" $\{\mathbf{x}_m\}_{m=1}^{M}$, where all $\mathbf{x}_m$ are 3D points (e.g., task-space points on the robot, semantic keypoints on objects, vertices that define arbitrary splines, surfaces, volumes or other geometric primitives, or any other information that should be inferred from visual input to control robot behavior). As defined in Sec.5.2, 5.3, let $\mathbb{A}(\mathbf{x_m})$ denote the binary random variable that $\mathbf{x_m}$ belongs to a certain part and let $\mathbb{R}(\mathbf{x_m})$ denote the metadata it has. Using all $\mathbf{x}_m$, $\mathbb{A}(\mathbf{x_m})$, and $\mathbb{R}(\mathbf{x_m})$ as optimization variables, suppose that we also transcribe an optimization loss $\ell(\mathbf{x}_{1:M}, \mathbb{A}(\mathbf{x}_{1:M}), \mathbb{R}(\mathbf{x}_{1:M}))$ capturing desirable outcomes and constraints involved in a task[2]. Note that $\mathbf{x}_{1:M}$ itself can be parametrized through any coordinate map (e.g., forward kinematics). To infer affordances, this optimization loss is treated as a probability distribution $\frac{1}{Z} e^{-\ell(\mathbf{x}_{1:M}, \mathbb{A}(\mathbf{x}_{1:M}), \mathbb{R}(\mathbf{x}_{1:M}))}$.[3] As defined in Sec.4 and Sec.5.3, different outcomes $\{\mathbb{A}_m\}_{m=1}^{M}$ and $\{\mathbb{R}_m\}_{m=1}^{M}$ for part and metadata assignments on the visual decision variables have

---

[2]We assume that the loss goes to $\infty$ when constraints are violated

[3]Note that configurations with a low loss are assigned a higher probability density, and as usual, the constant normalization factor $\frac{1}{Z}$ is ignored as it has no influence on inference (Thrun, 2002).

probability densities $\prod_{m=1}^{M} p[\mathbb{A}(\mathbf{x}_m) = \mathbb{A}_m]$ and $\prod_{m=1}^{M} p[\mathbb{R}(\mathbf{x}_m) = \mathbb{R}_m]$, respectively. These three distributions corresponding to the optimization loss, part assignments and metadata assignments are then multiplied together (with an independence assumption) to form a single probability distribution $p(\mathbf{x}_{1:M}, \mathbb{A}(\mathbf{x}_{1:M}), \mathbb{R}(\mathbf{x}_{1:M})))$ which captures how well any outcome $\mathbf{x}_{1:M}, \mathbb{A}(\mathbf{x}_{1:M}), \mathbb{R}(\mathbf{x}_{1:M})$ achieves the task-relevant affordance. We then infer likely outcomes from this distribution using Monte Carlo sampling routines to obtain a particle-filter representation (Thrun, 2002). As sampling from arbitrary and highly multi-modal distributions is very much an open problem in statistics (Gelman et al., 2013; Brooks et al., 2011), our Monte Carlo routines are inevitably task-specific and ad-hoc in nature. At this point, a sensible question to ask is why the optimization loss $\ell$ is treated as a probability distribution $\frac{1}{Z} e^{-\ell(\mathbf{x}_{1:M}, \mathbb{A}(\mathbf{x}_{1:M}), \mathbb{R}(\mathbf{x}_{1:M}))}$ rather than minimizing it directly. For an elaboration, please see Sec.A.4.1. If needed, please also see Sec.A.4.2 for a simple example to clarify and ground the abstract discussion provided here.

## 6 VISUAL MEMEX: APPLICATIONS TO GRASPING

We now describe how Vimex is applied to vision-based 6DOF grasping with a two-finger parallel-jaw gripper (Yan et al., 2018; Song et al., 2020; Sundermeyer et al., 2021; Du et al., 2021). First, we observe that such vision-based grasping scenarios are covered by two mutually exclusive subsets based on whether the grasp axis (i.e., the line that passes through the two gripper fingers) intersects the camera plane or is parallel to it. We call the former a "visible grasp" since one of the intended points of contact is always visible to the camera, and the latter an "occluded grasp" since both intended points of contact are not visible (this is sometimes called a top-down grasp (Mahler et al., 2017; Kleeberger et al., 2020; Tang et al., 2021; Zhao et al., 2021)). We define two separate affordances for these two cases. For the visible grasp, there is a single annotation set $\mathbb{A}_{\text{part}}$ that captures the task-relevant surface to be grasped (e.g., the handle of a pan or a drill). The metadata $\mathbb{R}_{\text{part}}$ attached to these annotations capture their antipodal distances, as defined and collected using the procedure described in Sec.4. The single decision variable $\mathbf{x}_c$ corresponds to the intended point of contact on the object surface, and is parametrized by the end-effector pose $\mathbf{T}_{\text{end}} \in SE(3)$. The optimization loss $\ell(\mathbf{x}_c, \mathbb{A}(\mathbf{x}_c), \mathbb{R}(\mathbf{x}_c))$ captures the following objectives and constraints:

- $\mathbf{x}_c$ should lie on the object surface within $m_{\text{obj}}$,
- The total probability mass $\int_V p[\mathbb{A}(\mathbf{x}) = \mathbb{A}_{\text{part}}] \, d\mathbf{x}$ inside a rectangular volume $V$ centered around $\mathbf{x}_c$ should be maximized,
- $|\hat{\mathbf{w}}^T \hat{\mathbf{n}}_c|$ should be maximized, where $\hat{\mathbf{w}}(\mathbf{T}_{\text{end}})$ is the direction of the line passing through the two gripper fingers and $\hat{\mathbf{n}}_c(\mathbf{x}_c)$ is the outwards surface normal on $\mathbf{x}_c$,
- $||\mathbf{x}_c - \frac{1}{2} \mathbb{R}(\mathbf{x}_c) \hat{\mathbf{n}}_c - \mathbf{x}_{\text{mid}}||$ should be minimized, where $\mathbf{x}_{\text{mid}}(\mathbf{T}_{\text{end}})$ is the midpoint of the gripper fingers and the overall term tries to center this midpoint half the antipodal distance away from $\mathbf{x}_c$ towards the inside of the object,
- The end-effector shouldn't intersect the visible points on the object surface within $m_{\text{obj}}$.

The associated Monte Carlo routine for inference first samples points $\mathbf{x}_c$ within $m_{\text{obj}}$, samples $\mathbf{x}_{\text{mid}}$ from a gaussian centered at $\mathbf{x}_c - \frac{1}{2} \mathbb{R}(\mathbf{x}_c) \hat{\mathbf{n}}_c$, samples a rotation angle $\alpha$ around the rotation axis $\hat{\mathbf{n}}_c$, and finally converts the rotation $\alpha, \hat{\mathbf{n}}_c$ and translation $\mathbf{x}_{\text{mid}}$ to $\mathbf{T}_{\text{end}} \in SE(3)$. The set of $(\mathbf{x}_c, \mathbf{T}_{\text{end}})$ sampled this way are treated as particles of a particle filter, their weights are computed using $p(\mathbf{x}_c, \mathbb{A}(\mathbf{x}_c), \mathbb{R}(\mathbf{x}_c))$ and then normalized. For the occluded grasp, the same annotation set $\mathbb{A}_{\text{part}}$ as the visible grasp is used. This time, no metadata is attached to these annotations. There is again a single decision variable $\mathbf{x}_{\text{mid}}$ parametrized by $\mathbf{T}_{\text{end}} \in SE(3)$ that corresponds to the midpoint of the gripper fingers. The optimization loss $\ell(\mathbf{x}_{\text{mid}}, \mathbb{A}(\mathbf{x}_{\text{mid}}))$ captures the following:

- $\mathbf{x}_{\text{mid}}$ should lie on the object surface within $m_{\text{obj}}$,
- The total probability mass $\int_V p[\mathbb{A}(\mathbf{x}) = \mathbb{A}_{\text{part}}] \, d\mathbf{x}$ inside a rectangular volume $V$ centered around $\mathbf{x}_{\text{mid}}$ should be maximized,
- The minimum distance between all points on the end-effector and all occluded points within $V_{\text{obj}}$ (i.e., $f_{\text{obj}}(\mathbf{x}) = \mathbf{z}_{\text{occ}}$ from Sec.5.2) should be maximized,
- The end-effector shouldn't intersect the visible or occluded points within $V_{\text{obj}}$ (i.e., all points on the end-effector should lie in empty space).

| | Any Top-K | | | K-NN | | |
|---|---|---|---|---|---|---|
| | Average | Chair | Elephant | Average | Chair | Elephant |
| N=1 | 73.6 ± 8.3 | 53.6 ± 3.0 | 96.8 ± 7.1 | 58.6 ± 4.8 | 50.4 ± 0.6 | 70.9 ± 8.7 |
| N=5 | 87.4 ± 2.8 | 63.4 ± 4.1 | 99.1 ± 0.2 | 67.5 ± 3.0 | 51.6 ± 1.3 | 95.3 ± 5.6 |
| N=10 | 91.4 ± 1.6 | 70.1 ± 3.3 | 99.2 ± 0.2 | 72.2 ± 2.4 | 52.9 ± 1.3 | 97.8 ± 1.0 |
| N=25 | 94.7 ± 0.8 | 79.8 ± 2.0 | 99.3 ± 0.2 | 78.1 ± 1.5 | 56.5 ± 1.5 | 98.4 ± 0.2 |
| N=50 | 96.3 ± 0.6 | 85.2 ± 1.4 | 99.5 ± 0.2 | 82.0 ± 1.1 | 59.8 ± 1.2 | 98.7 ± 0.2 |
| N=100 | 97.2 ± 0.4 | 89. 7 ± 1.0 | 99.5 ± 0.1 | 85.5 ± 0.8 | 63.9 ± 0.9 | 99.4 ± 0.1 |

**Table 1:** Classification accuracies on the COCO dataset for nearest-neighbor based object recognition, with (i.e., Any Top-K) and without (i.e., K-NN) the proposed approach to counter the imbalance between the sizes of $(\mathbb{D}_{pos}, \mathbb{D}_{neg})$. $N$ is the number of positive exemplars, Average is the mean accuracy across all categories, while Chair and Elephant are the hardest and easiest (i.e., most and least visually diverse) categories respectively (please see Table.3 for complete results on all other categories). Standard deviations are computed over 1000 random samples of the $N$ exemplars. As can be seen, default K-NN performs poorly, and introducing the any top-K approach increases the classification accuracy above %90 using as few as 10 exemplars, thus addressing the functional and practical requirements previously identified for a task description framework.

| | Average | Bird Head | Bird Body | Bird Wing | Bird Foot | Bird Tail |
|---|---|---|---|---|---|---|
| N=2 , S=14 | 82.9 | 66.2 | 63.5 | 85.3 | 68.6 | 66.4 |
| N=2 , S=7 | 88.7 | 81.4 | 71.1 | 90.8 | 89.2 | 69.6 |
| N=5 , S=7 | 91.6 | 90.8 | 85.3 | 90.5 | 86.8 | 89.7 |

**Table 2:** Part localization success rate on the PartImageNet dataset. $N$ is the number of annotated images, $S$ is the stride used for the spatial descriptor grid (i.e., lower means denser), and Average is the mean success rate across all part categories. Also shown are five parts of the bird supercategory as they provide a good illustrative example of the common trends across all categories (please see Table.4 for complete results on all other part categories). It can be seen that small parts (e.g., Bird Foot) mainly benefit from an increase in spatial descriptor density while parts with less clear-cut definitions (e.g., more visual variety, like Bird Tail) mainly benefit from an increase in the number of exemplars. Overall, $\sim$ 5 annotations are sufficient for a success rate above %90, addressing the functional and practical requirements previously identified for a task description framework.

The associated Monte Carlo routine for inference first samples points $\mathbf{x}_{mid}$ within $m_{obj}$, samples a rotation axis $\hat{\mathbf{w}}$ such that $\hat{\mathbf{w}}^T \hat{\mathbf{n}}_{mid}$ is positive, samples a rotation angle $\alpha$, and finally converts the rotation $\alpha$, $\hat{\mathbf{w}}$ and translation $\mathbf{x}_{mid}$ to $\mathbf{T}_{end} \in SE(3)$. These samples $(\mathbf{x}_{mid}, \mathbf{T}_{end})$ and their weights $p(\mathbf{x}_{mid}, \mathbb{A}(\mathbf{x}_{mid}))$ are again represented as a particle filter. The two particle filters that represent 6DOF pose distributions for the visible and occluded grasps are then mixed together and forwarded to the downstream planning and control pipeline to generate behavior.

## 7 EXPERIMENTS

This section presents experimental evaluations for the three main capacities provided by Vimex: object recognition, part localization, and inferring affordances.

**Object-recognition:** Evaluations use the train-subset of the COCO dataset (Lin et al., 2014), by first converting it to an image classification dataset. To achieve this, for all object instance masks $\{m_i\}$ in every image, the background is removed, a tight square is cropped around the mask, and resized to the standard resolution, resulting in an image where a single object is present over a black background. The reason we opt for this evaluation method, rather than using COCO as is and running an instance segmentation evaluation, is because the proposed method in Sec.5.1 solely focuses on object classification after an off-the-shelf class agnostic segmentation method provides the masks $\{m_i\}$. Table.1 shows the resulting classification accuracies for K-NN without balancing and for the proposed any top-K balancing method. As can be seen, the default K-NN classifier performs poorly, and introducing the any top-K approach increases the classification accuracy above %90 for most categories using as few as 10 exemplars to represent a category. Such few numbers of exemplars being sufficient for good performance suggests that the first step of isolating and standardizing object

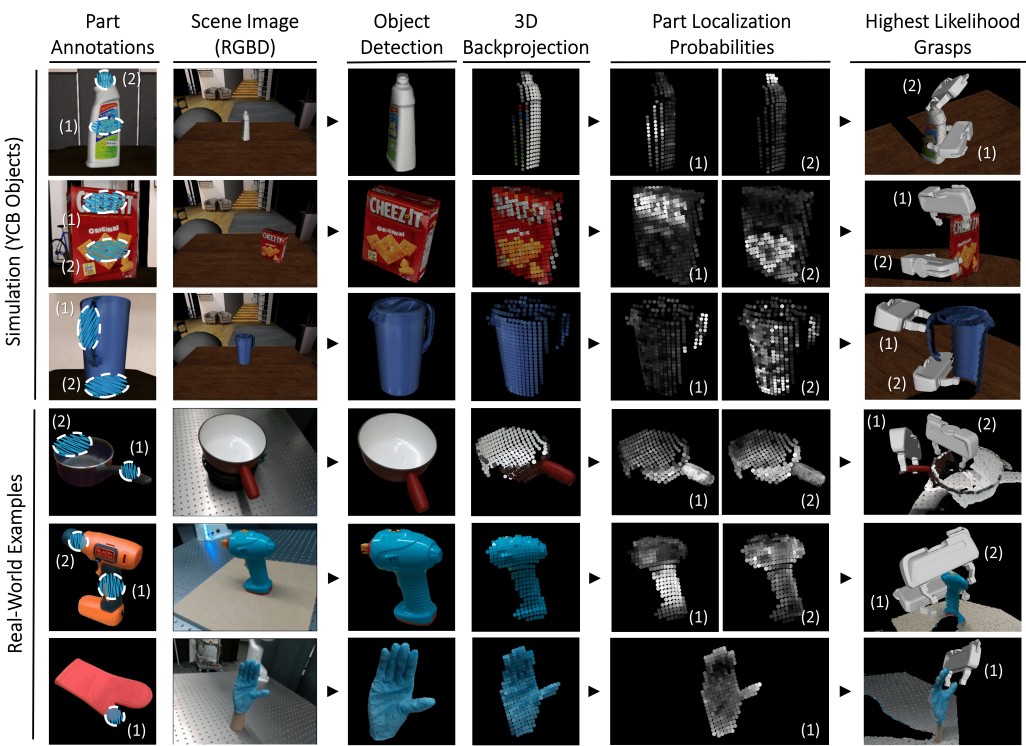

**Figure 2:** A visualization of the complete Vimex pipeline and the respective results of object recognition, part localization, and the inference of grasp affordances, on 3 simulated and 3 real-world examples. As can be seen, part localization generalizes to different poses and inter-category appearance variations to produce well-behaved distributions, resulting in antipodal grasps that cover the correct object parts without collisions.

instances (i.e., by background removal, cropping, and resizing) makes the subsequent classification much easier (e.g., compared to the Imagenet K-NN evaluations presented in (Oquab et al., 2023)).

**Part Localization:** Evaluations use the train-subset of the PartImageNet dataset (He et al., 2022) with the following Monte Carlo procedure. A single Monte Carlo trial for any part category starts with randomly sampling $N$ annotation images $\{I_{\text{ann},i}\}_{i=1}^N$ and a target image $I_{\text{target}}$ that all contain the part, together with their ground truth part masks $\{m_{\text{ann},i}\}_{i=1}^N$ and $m_{\text{target}}$. All spatial descriptors within $\{m_{ann,i}\}_{i=1}^N$ are pooled to form the annotation set $\mathbb{A}_{part}$. The distribution $p[\mathbb{A}(\mathbf{p}) = \mathbb{A}_{\text{part}}]$ is then computed over all spatial descriptors in $I_{\text{target}}$, and the pixel coordinate $\mathbf{p}$ with the highest probability mass is selected. If $\mathbf{p}$ lies within $m_{\text{target}}$, it means that the target part is localized successfully. For each part category, 1000 such Monte Carlo trials are run. As shown in Table.2, the resulting part localization success rates increase well above %90 using as few as 5 annotations to represent a part, validating the efficacy of the proposed approach.

**Inferring Affordances:** There are two sets of evaluations: in simulation and in the real-world. For simulated experiments, a table-top manipulation station in a private dwelling setting is set with complete textures and realistic rendering, using the Drake simulation environment (Tedrake & the Drake Development Team, 2019) with assets from ReplicaCAD (Szot et al., 2021). This setup contains 3 different objects (i.e., cracker-box, pitcher, bleach) placed in 4 different poses on a table, resulting in 12 scenes in total. All objects are defined using 5 RGBD images captured away from the scene, and there are two different part annotations per object. Given an RGBD image, the task is to generate antipodal grasps on the object around each of the two part annotations. Fig.2 visualizes the highest likelihood grasps and the intermediate steps for a subset of scenes for brevity (please see Sec.A.4.1 and the supplementary material for the full results). For all 12 scenes, the inferred grasps are antipodal, free of collisions, and cover the correct parts. For real-world experiments, a similar table-top manipulation station is set up containing 3 different object categories with 2 instances per category (i.e., 6 objects in total: black/red pan, red/blue drill, pink/blue glove). All objects are

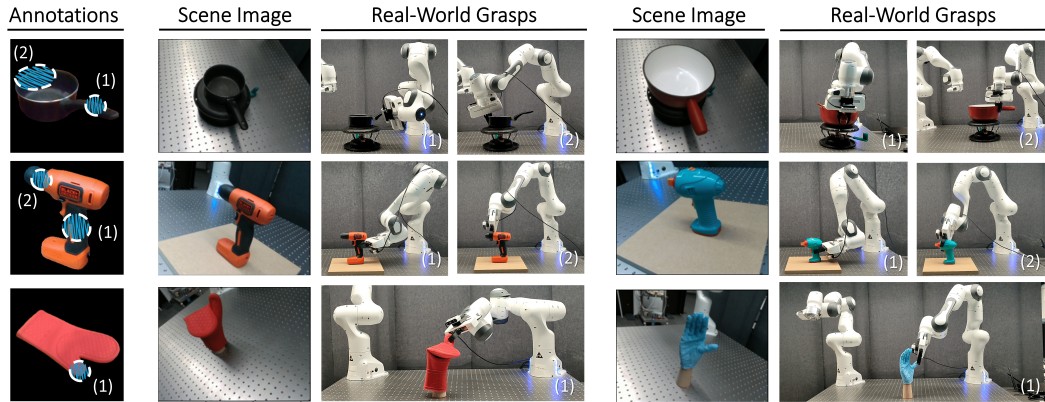

**Figure 3:** Real-world evaluations on 10 different grasping scenarios involving 3 object categories. As can be seen, a single annotated image provides a sufficient task description that generalizes to different object poses as well as to category-level variations in shape and appearance.

defined using 10 RGBD images captured away from the scene, and there are 5 part annotations in total (i.e., pan handle/side, drill handle/top, glove thumb). To test generalization, annotations solely come from the images of one instance (i.e., black pan, red drill, pink glove) while the task is to generate grasps given RGBD images of both instances. Fig.2 visualizes the highest likelihood grasps and the intermediate steps for a subset of the scenes, while Fig.3 shows a robot hand executing these grasps for all scenes (please see Sec.A.4.1 and the supplementary material for videos). Again, it can be seen that all 10 grasps are antipodal, free of collisions, and cover the correct parts.

## 8 CONCLUSION

The main focus of this paper is the task-description problem in robotics, particularly in settings where the following three considerations apply: i) the robot will be deployed in an a priori unknown and uninstrumented environment; ii) it will be expected to perform an a priori unknown, open-ended, and ever-changing set of tasks in that environment; iii) every such task will define its own notions of objects, parts, and affordances. With these considerations in mind, we proposed a framework named Vimex, which allows a user to describe arbitrary objects, parts, and affordances involved in a vision-based task to a robot. This description process is intended to be quick (i.e., no need to train a neural network), easy (i.e., no specialized equipment is necessary other than a consumer camera), and efficient (i.e., ~10 RGB images with scribble annotations are sufficient). Components of Vimex were evaluated on COCO, PartImageNet, and on simulated and real-world robotic grasping scenarios. The main direction for future work is to expand the decision variable and metadata types involved in Vimex such that given RGBD images from a scene, the association and metadata retrieval processes described here can be employed to assemble together a complete intuitive-physics simulation (Battaglia et al., 2013) for that scene. Such a simulation can then be utilized as is by existing planners (Cohn et al., 2023; Toussaint et al., 2022; Garrett et al., 2021) to enable vision-based decision-making in a POMDP setting (Kaelbling et al., 1998; Botvinick & Toussaint, 2012).

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

# A    APPENDIX

## A.1    RELATED WORK (EXPANDED)

### A.1.1    CONTROL FROM PIXELS

Reinforcement learning (RL)  (Sutton & Barto, 2018; Schulman et al., 2017; Haarnoja et al., 2018) and behavior cloning (BC)  (Billard et al., 2008; Chi et al., 2023; Xie et al., 2023) algorithms excel at learning a single complex task and can incorporate images as part of their state representations. But on their own, they do not provide an efficient interface to describe, modify, or combine open-ended sets of different tasks on the fly (e.g., without implementing a reward/reset mechanism or collecting $\sim$100 demonstrations to train a model for a few hours). Therefore it may be more suitable to treat them as parametrizable (i.e., goal-conditioned) skills within a higher task-level planning framework (Konidaris et al., 2018; 2012; Sutton et al., 1999). Affordances inferred by Vimex can potentially be employed as an interface for such a task-level planner to pick and parametrize RL/BC based skills (e.g., see  Curtis et al. (2022)), highlighting a synergy between Vimex, Task and Motion Planning (TAMP)  (Toussaint, 2015; Garrett et al., 2021; Toussaint et al., 2022), and RL/BC.

Another related approach is using vision-language-action models (VLAs)  (Brohan et al., 2023; Driess et al., 2023; Ahn et al., 2022; Shridhar et al., 2022). These models represent images, text, and actions using the same format (e.g., explicitly as input-output tokens of a transformer or implicitly in the joint representation space of multiple encoders), and then capture their co-occurrence statistics with a probabilistic model trained over a very large (e.g., internet-scale) dataset. Generating samples from this distribution conditioned on the robot's current context and a task description (represented in the same format as images, text, and actions) in turn can produce a wide range of sensible behavior. While this approach shows very impressive results, these capacities are mostly emergent and not interpretable. In comparison, affordances in Vimex are inferred from probability densities that are built piece-by-piece from exemplars and optimization losses, making the overall framework more interpretable and easy to debug, understand, and control. That being said, inference and sampling on these arbitrarily complex probability densities is a challenging problem, and the statistical contingencies captured by VLAs can help initialize and guide this process to make it faster and more effective (see  Fang et al. (2023); Yang et al. (2023) for relevant ideas).

### A.1.2    IMAGE CLASSIFICATION AND SEGMENTATION

A standard approach to image classification  (Krizhevsky et al., 2012; He et al., 2016; Dosovitskiy et al., 2020) or segmentation  (He et al., 2017; Cheng et al., 2021; Jain et al., 2023) is to train a model on the fixed class definitions provided by a dataset  (Deng et al., 2009; Zhou et al., 2019).  This approach may not be the most suitable task-description method for robotics applications, because of the observations and requirements outlined in Sec.1.  Another approach is that of foundation models  (Bommasani et al., 2021) trained on internet scale data that can be prompted  (Gu et al., 2023) to adapt them for new tasks without modifying their weights. Of particular importance to us are  (Oquab et al., 2023; Kirillov et al., 2023), as the statistics they capture about objectness and

co-occurance probabilities between image regions (see Lowe (1985)) are central to the way Vimex localizes object parts and retrieves the associated metadata.

## A.2 PRELIMINARIES

### A.2.1 GLOBAL AND SPATIAL DESCRIPTORS OF DINOv2

The inner-product $\mathbf{z}_1^T \mathbf{z}_2$ between the spatial descriptors of DINOv2 implements a semantically meaningful nearest-neighbors similarity metric for image retrieval and patch matching. The training process can provide a qualitative explanation for this observation. It randomly samples patches from a batch of images retrieved from internet-scale data. Then, it minimizes a loss function that causes the inner products between descriptors of patches that come from the same image to be closer compared to those that come from different images. This means that at the end of training, $\mathbf{z}_1^T \mathbf{z}_2$ acts similar to an unnormalized probability density that captures the likelihood of two patches being co-observed within the same image, bringing to mind the "probability of non-accidental co-occurance" described by Lowe (1985) and the aspect-graphs described by Koenderink & Van Doorn (1979). Since analogous sets of points on two different physical entities from the same semantic category have similar co-observation probabilities, the inner-product distributions within these two sets also end up being similar. And because the combined effects of internet-scale data, limited representation capacity, and training regularization create an incentive for compression (Tishby & Zaslavsky, 2015), these two sets of descriptors end up being packed closely rather than redundantly replicating their similar metric structure on different parts of the descriptor space.

## A.3 VISUAL MEMEX: USER INPUTS FOR TASK-DESCRIPTION

### A.3.1 MEASURING ANTIPODAL DISTANCES

The user first retrieves a physical instance of an object (e.g., their personal cup, or a strawberry from the field). They then capture multiple RGB images of it from different viewpoints, which are automatically converted to a metrically accurate 3D mesh of the object through a standard pipeline (Schonberger & Frahm, 2016; Cignoni et al., 2008). Every vertex $v_i$ on this object mesh is associated with a spatial descriptor $\mathbf{z}_i$ by projecting it to the nearest image and computing the spatial descriptor at this projection coordinate. Finally, the antipodal distance $d_i$ is computed by following the surface normal $n_i$ inwards starting from vertex $v_i$ until the first intersection with the object surface on the other side. This way, a record $(z_i, \mathbb{R}_i = \{d_i\})$ is constructed from every vertex $v_i$. This process can be iterated to scan an arbitrary number of objects (i.e. to improve category-level generalization), and the associated records are pooled together to form the memory $\mathcal{M} = \{(z_i, \mathbb{R}_i)\}$.

## A.4 VISUAL MEMEX: INFERENCE

### A.4.1 INFERRING AFFORDANCES

First, note that the minimum of the optimization loss directly corresponds to a maximum likelihood estimate for the latter density. Still, explicitly representing the entire distribution as a particle filter rather than a single point estimate has the advantage of allowing a downstream planning and control pipeline to reason about uncertainty, particularly when the loss is multimodal and different modes of behavior are valid. Furthermore, it is often the case that the loss function we transcribe is not a perfect description of the behavior we want, or we do not have access to ground-truth objectives or dynamics but instead work with surrogates or approximate models. In such cases, sensible but not necessarily globally optimal samples are still valuable as they can be tested in the real-world to debug and improve all components (e.g., task-description input from the user, the optimization loss definition) through trial and error. For more elaboration on these motivations as well as theoretical connections between sampling and optimization, see (Cheng, 2020).

### A.4.2 INFERRING AFFORDANCES: A CONCRETE EXAMPLE

To give a concrete example, consider a robot that should spread jam over sliced bread. It first needs to place a knife on the bread over a region with jam, and then move it on a linear trajectory towards a region without jam. To describe this task using Vimex, the user manually spreads jam on one

half of a bread in their kitchen, takes an image of this example demonstration, and provides scribble annotations $\{\mathbb{A}_{\text{jam},+}, \mathbb{A}_{\text{jam},-}\}$ on two regions with and without jam. There is no metadata attached to these regions for this simple example. This single demonstration is sufficient for later localizing $\{\mathbb{A}_{\text{jam},+}, \mathbb{A}_{\text{jam},-}\}$ on any slice of bread in anyone's kitchen. For inference, the visual decision variables $\{\mathbf{x}_m\}_{m=1}^3$ correspond to the start, middle, and end points of a 3D line, and the optimization loss $\ell(\mathbf{x}_{1:3}, \mathbb{A}(\mathbf{x}_{1:3}))$ captures the following objectives and constraints: i) $\mathbb{A}(\mathbf{x}_1) = \mathbb{A}_{\text{jam},+}$ should hold, ii) $\mathbb{A}(\mathbf{x}_3) = \mathbb{A}_{\text{jam},-}$ should hold, iii) all $\mathbf{x}_{1:3}$ should lie on the object surface within $m_{\text{obj}}$, iv) $\|\mathbf{x}_1 - \mathbf{x}_3\|$ should be maximized. The associated Monte Carlo routine for inference simply samples random pairs of points within $m_{\text{obj}}$, computes $\mathbf{x}_{1:3}$ for the corresponding line segment, treats all sampled $\{\mathbf{x}_{1:3}\}$ as particles of a particle filter, evaluates $p(\mathbf{x}_{1:3}, \mathbb{A}(\mathbf{x}_{1:3}))$ for each particle to compute their weights, and normalizes the sum of their weights to be 1. The single line with the highest weight can then be followed by the end effector using an appropriate controller, or the entire particle filter can be used by a more complex pipeline that makes use of uncertainty.

## A.5 EXPERIMENTS

### A.5.1 INFERRING GRASP AFFORDANCES

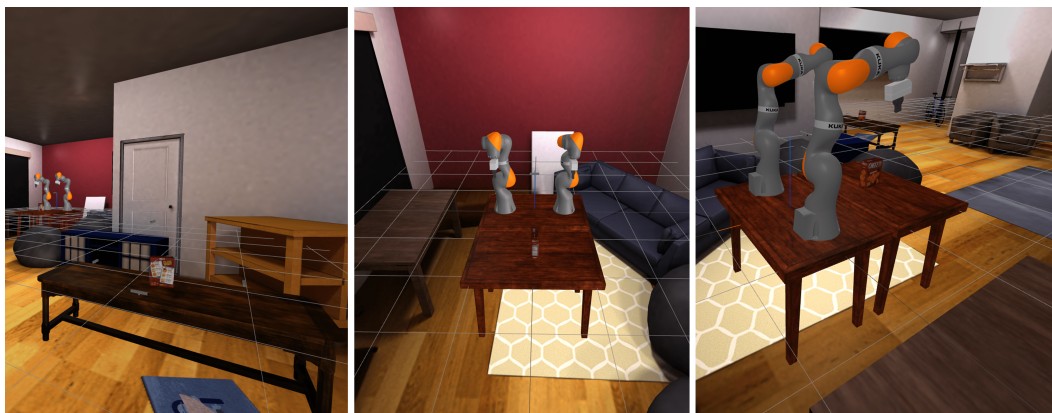

**Figure 4:** The simulation environment is built using Drake and ReplicaCAD. It contains a manipulation station placed in an apartment and a scanning table.

Fig.4 shows images of the simulation environment. For grasp outputs of all 12 simulated scenes and grasping videos of all 10 real-world scenes, please see the folders "/experiments/sim" and "/experiments/real_world" in the supplementary material.

### A.5.2 COMPLETE EVALUATIONS ON COCO AND PARTIMAGENET

Please find below the tables that list complete accuracy evaluations for all COCO categories and complete part localization success rate evaluations for all PartImageNet categories.

|  | N=1 | N=5 | N=10 | N=25 | N=50 | N=100 |
|---|---|---|---|---|---|---|
| person | 65.5 ± 9.2 | 85.6 ± 3.9 | 91.5 ± 1.4 | 95.3 ± 0.7 | 96.5 ± 0.6 | 97.0 ± 0.4 |
| bicycle | 85.0 ± 11.0 | 96.3 ± 0.8 | 97.1 ± 0.5 | 97.8 ± 0.4 | 98.3 ± 0.5 | 98.4 ± 0.5 |
| car | 59.6 ± 6.6 | 76.9 ± 3.5 | 83.8 ± 2.2 | 90.3 ± 1.4 | 93.7 ± 0.7 | 95.6 ± 0.5 |
| motorcycle | 90.7 ± 9.5 | 97.7 ± 0.7 | 98.3 ± 0.5 | 98.8 ± 0.3 | 99.0 ± 0.2 | 99.1 ± 0.2 |
| airplane | 63.9 ± 4.5 | 89.8 ± 3.8 | 96.4 ± 1.8 | 98.7 ± 0.5 | 99.1 ± 0.2 | 99.2 ± 0.2 |
| bus | 77.5 ± 10.9 | 93.7 ± 1.7 | 95.6 ± 1.2 | 97.3 ± 0.5 | 98.0 ± 0.3 | 98.4 ± 0.2 |
| train | 72.3 ± 9.8 | 92.4 ± 3.4 | 96.2 ± 1.4 | 98.1 ± 0.5 | 98.7 ± 0.3 | 99.1 ± 0.2 |
| truck | 59.3 ± 5.0 | 79.0 ± 3.5 | 87.1 ± 2.5 | 93.9 ± 0.9 | 95.9 ± 0.4 | 96.9 ± 0.3 |
| boat | 65.3 ± 8.6 | 84.7 ± 3.2 | 90.1 ± 1.4 | 93.4 ± 0.9 | 95.1 ± 0.7 | 96.3 ± 0.6 |
| fire hydrant | 98.9 ± 4.9 | 99.6 ± 0.1 | 99.6 ± 0.2 | 99.6 ± 0.1 | 99.5 ± 0.2 | 99.5 ± 0.2 |
| stop sign | 96.7 ± 6.8 | 98.8 ± 0.5 | 99.1 ± 0.5 | 99.4 ± 0.4 | 99.7 ± 0.3 | 99.8 ± 0.2 |
| bench | 73.2 ± 11.3 | 85.7 ± 2.7 | 88.8 ± 1.5 | 92.3 ± 1.3 | 94.4 ± 0.8 | 95.8 ± 0.5 |
| bird | 54.8 ± 2.7 | 68.4 ± 3.7 | 79.4 ± 2.9 | 91.5 ± 1.7 | 96.4 ± 1.1 | 98.3 ± 0.7 |
| cat | 64.2 ± 5.7 | 89.7 ± 4.4 | 96.4 ± 1.5 | 98.6 ± 0.4 | 98.9 ± 0.2 | 99.0 ± 0.2 |
| dog | 59.1 ± 4.4 | 79.6 ± 2.9 | 88.4 ± 2.1 | 95.9 ± 0.9 | 97.8 ± 0.4 | 98.5 ± 0.2 |
| horse | 84.1 ± 8.1 | 97.8 ± 1.5 | 98.9 ± 0.4 | 99.2 ± 0.1 | 99.3 ± 0.1 | 99.3 ± 0.2 |
| sheep | 83.2 ± 13.5 | 93.9 ± 1.2 | 95.6 ± 1.2 | 97.0 ± 1.1 | 97.9 ± 0.7 | 98.4 ± 0.4 |
| cow | 92.3 ± 9.5 | 97.5 ± 0.5 | 98.1 ± 0.4 | 98.4 ± 0.3 | 98.7 ± 0.2 | 98.7 ± 0.3 |
| elephant | 96.8 ± 7.1 | 99.1 ± 0.2 | 99.2 ± 0.2 | 99.3 ± 0.2 | 99.5 ± 0.2 | 99.5 ± 0.1 |
| bear | 85.7 ± 9.4 | 96.5 ± 1.2 | 97.9 ± 0.7 | 98.5 ± 0.4 | 99.0 ± 0.3 | 99.3 ± 0.3 |
| zebra | 98.9 ± 0.9 | 99.6 ± 0.2 | 99.8 ± 0.1 | 99.9 ± 0.1 | 99.9 ± 0.1 | 99.9 ± 0.1 |
| giraffe | 98.5 ± 6.5 | 99.8 ± 0.1 | 99.9 ± 0.1 | 99.8 ± 0.1 | 99.8 ± 0.1 | 99.8 ± 0.1 |
| umbrella | 92.5 ± 11.1 | 97.4 ± 0.3 | 97.8 ± 0.3 | 98.1 ± 0.2 | 98.3 ± 0.2 | 98.6 ± 0.3 |
| suitcase | 65.8 ± 9.6 | 82.3 ± 3.0 | 86.7 ± 1.8 | 90.7 ± 1.1 | 93.1 ± 1.0 | 94.9 ± 0.7 |
| cup | 59.9 ± 5.5 | 80.9 ± 5.7 | 88.4 ± 2.6 | 93.3 ± 1.0 | 95.0 ± 0.8 | 95.9 ± 0.7 |
| bowl | 54.1 ± 3.6 | 65.7 ± 4.4 | 73.8 ± 3.7 | 84.4 ± 1.4 | 89.3 ± 1.0 | 92.2 ± 0.7 |
| banana | 76.0 ± 13.9 | 92.2 ± 2.7 | 94.2 ± 1.7 | 97.0 ± 1.1 | 98.1 ± 0.7 | 98.5 ± 0.4 |
| sandwich | 56.6 ± 5.0 | 72.5 ± 5.0 | 81.7 ± 4.1 | 89.8 ± 1.8 | 93.3 ± 1.1 | 95.5 ± 0.7 |
| orange | 82.6 ± 11.2 | 96.0 ± 1.6 | 97.5 ± 0.7 | 98.2 ± 0.6 | 98.6 ± 0.5 | 99.0 ± 0.6 |
| broccoli | 82.6 ± 12.3 | 94.8 ± 2.4 | 96.8 ± 1.4 | 98.2 ± 0.6 | 98.7 ± 0.5 | 99.0 ± 0.5 |
| hot dog | 77.1 ± 16.9 | 93.5 ± 2.0 | 94.7 ± 0.6 | 95.8 ± 0.6 | 96.5 ± 0.6 | 97.3 ± 0.7 |
| pizza | 60.3 ± 7.1 | 84.1 ± 5.7 | 92.1 ± 3.4 | 97.5 ± 1.0 | 98.3 ± 0.2 | 98.1 ± 0.2 |
| donut | 68.4 ± 12.2 | 86.8 ± 4.2 | 91.1 ± 1.8 | 95.2 ± 1.2 | 97.3 ± 0.8 | 97.9 ± 0.6 |
| cake | 54.5 ± 3.8 | 66.6 ± 5.1 | 75.5 ± 4.0 | 85.7 ± 1.7 | 90.4 ± 1.1 | 93.3 ± 0.7 |
| chair | 53.6 ± 3.0 | 63.4 ± 4.1 | 70.1 ± 3.3 | 79.8 ± 2.0 | 85.2 ± 1.4 | 89.7 ± 1.0 |
| couch | 60.1 ± 8.0 | 76.2 ± 4.3 | 81.6 ± 2.3 | 87.3 ± 1.7 | 90.9 ± 1.2 | 93.1 ± 0.6 |
| potted plant | 64.9 ± 9.6 | 86.6 ± 4.5 | 92.4 ± 1.9 | 96.2 ± 0.8 | 97.4 ± 0.5 | 98.0 ± 0.5 |
| bed | 65.2 ± 10.8 | 81.3 ± 4.2 | 86.2 ± 2.2 | 90.0 ± 1.2 | 92.4 ± 0.7 | 94.0 ± 0.4 |
| dining table | 52.1 ± 1.7 | 59.2 ± 3.0 | 66.1 ± 2.4 | 76.9 ± 2.0 | 84.5 ± 1.4 | 89.4 ± 1.0 |
| toilet | 95.1 ± 2.4 | 97.4 ± 0.5 | 97.8 ± 0.5 | 98.6 ± 0.3 | 98.9 ± 0.3 | 99.0 ± 0.2 |
| tv | 61.3 ± 7.2 | 79.3 ± 5.8 | 87.1 ± 2.7 | 92.0 ± 0.8 | 94.3 ± 0.6 | 95.9 ± 0.5 |
| laptop | 74.5 ± 11.6 | 90.5 ± 3.1 | 93.9 ± 1.1 | 96.1 ± 0.6 | 97.0 ± 0.3 | 97.4 ± 0.3 |
| keyboard | 80.2 ± 11.6 | 95.6 ± 2.9 | 97.5 ± 1.7 | 98.9 ± 0.4 | 99.2 ± 0.4 | 99.3 ± 0.3 |
| cell phone | 81.1 ± 11.0 | 94.2 ± 1.5 | 95.9 ± 1.1 | 97.2 ± 0.6 | 97.8 ± 0.5 | 97.9 ± 0.7 |
| oven | 64.4 ± 9.3 | 82.9 ± 4.6 | 87.9 ± 1.8 | 91.7 ± 1.1 | 93.9 ± 0.9 | 95.4 ± 0.7 |
| sink | 69.0 ± 10.8 | 85.0 ± 2.7 | 88.1 ± 1.5 | 91.0 ± 1.1 | 92.5 ± 1.1 | 93.9 ± 1.3 |
| refrigerator | 72.8 ± 10.1 | 94.1 ± 3.7 | 96.2 ± 1.7 | 97.5 ± 0.4 | 98.1 ± 0.3 | 98.3 ± 0.2 |
| book | 60.6 ± 7.5 | 80.7 ± 4.8 | 86.2 ± 1.9 | 90.4 ± 0.9 | 92.2 ± 0.7 | 93.7 ± 0.7 |
| clock | 88.4 ± 11.7 | 96.5 ± 0.6 | 97.1 ± 0.8 | 97.9 ± 0.9 | 98.4 ± 0.8 | 98.7 ± 0.7 |
| vase | 67.2 ± 7.6 | 85.7 ± 4.5 | 92.2 ± 2.6 | 96.3 ± 0.8 | 97.4 ± 0.7 | 98.1 ± 0.6 |
| teddy bear | 88.3 ± 13.5 | 96.9 ± 0.5 | 97.3 ± 0.3 | 97.8 ± 0.3 | 98.0 ± 0.3 | 98.3 ± 0.3 |

**Table 3:** Classification accuracies on all categories of the COCO dataset for nearest-neighbor based object recognition, using the proposed Any Top-K approach for balancing with K=100.

|  | N=2 , S=14 | N=2 , S=7 | N=5 , S=7 |
|---|---|---|---|
| Quadruped Head | % 82 | % 93 | % 94 |
| Quadruped Body | % 86 | % 92 | % 93 |
| Quadruped Foot | % 85 | % 93 | % 96 |
| Quadruped Tail | % 73 | % 82 | % 85 |
| Biped Head | % 86 | % 87 | % 90 |
| Biped Body | % 65 | % 77 | % 81 |
| Biped Hand | % 82 | % 87 | % 87 |
| Biped Foot | % 80 | % 81 | % 85 |
| Biped Tail | % 83 | % 84 | % 92 |
| Fish Head | % 97 | % 100 | % 99 |
| Fish Body | % 96 | % 97 | % 98 |
| Fish Fin | % 87 | % 89 | % 92 |
| Fish Tail | % 94 | % 97 | % 93 |
| Bird Head | % 66 | % 81 | % 91 |
| Bird Body | % 64 | % 71 | % 85 |
| Bird Wing | % 85 | % 91 | % 91 |
| Bird Foot | % 69 | % 89 | % 87 |
| Bird Tail | % 66 | % 70 | % 90 |
| Snake Head | % 86 | % 90 | % 93 |
| Snake Body | % 98 | % 100 | % 98 |
| Reptile Head | % 88 | % 94 | % 99 |
| Reptile Body | % 91 | % 97 | % 98 |
| Reptile Foot | % 91 | % 99 | % 93 |
| Reptile Tail | % 83 | % 89 | % 96 |
| Car Body | % 99 | % 100 | % 100 |
| Car Tier | % 90 | % 98 | % 98 |
| Car Side Mirror | % 44 | % 54 | % 60 |
| Bicycle Body | % 89 | % 90 | % 90 |
| Bicycle Head | % 90 | % 91 | % 94 |
| Bicycle Seat | % 66 | % 82 | % 85 |
| Bicycle Tier | % 96 | % 98 | % 98 |
| Boat Body | % 99 | % 99 | % 100 |
| Boat Sail | % 97 | % 96 | % 100 |
| Aeroplane Head | % 69 | % 90 | % 99 |
| Aeroplane Body | % 91 | % 93 | % 98 |
| Aeroplane Engine | % 66 | % 72 | % 79 |
| Aeroplane Wing | % 78 | % 77 | % 76 |
| Aeroplane Tail | % 92 | % 93 | % 91 |
| Bottle Mouth | % 79 | % 88 | % 92 |
| Bottle Body | % 93 | % 99 | % 99 |

**Table 4:** Part localization success rates for all categories of the PartImageNet dataset.

