# OpenReview forum: "VIMEX: A Memory-Centered Task Description Framework for Vision-Based Robotics"
_ICLR.cc/2024/Conference — ICLR 2024 Conference Withdrawn Submission_

### Official Review · Reviewer_zCUf · 2023-10-26

**Soundness:** 2 fair
**Presentation:** 3 good
**Contribution:** 2 fair
**Rating:** 3
**Confidence:** 4

**Summary:**

Versatile robots should be able to operate in a variety of environments without any robot-specific infrastructure, and it should be possible to teach them diverse manipulation tasks efficiently and with ease. There is no one-size-fits-all world model; useful notions of objects, parts and affordances will differ between tasks. This paper proposes Vimex, a task specification and representation framework that allows non-expert users to communicate new tasks to a robot in terms of objects, parts, and grasp affordances. Vimex leverages DINOv2 for visual feature extraction, and Segment Anything for class-agnostic segmentation. It contributes methods for object recognition from images, part localization in 3D, and the prediction of grasp affordances for visible and invisible contact locations. Vimex' performance is evaluated in terms of these three core capabilities.

**Strengths:**

This paper addresses a problem of high practical importance. The ability of non-expert users to teach robots diverse tasks is essential in practice but totally unsolved. This work formulates the key insight that different tasks require different notions of objects, parts, and affordances, and places this insight center stage by striving to make exactly these three concepts easy to define for the user. Moreover, this work proposes practical solutions for turning these user-defined concepts into robot capabilities, leveraging existing, modern technologies. The performance of Vimex is evaluated on these three capabilities. Another nice feature is the ability to include arbitrary metadata and leverage it for robot inference.

The paper is pretty well written and mostly clear.

**Weaknesses:**

- The proposed framework employs a variety of methods and combines them together (DINOv2, SAM, Monte-Carlo sampling, particle representations, and specific methods for part localization and affordance detection). Each choice of method, by itself, is well motivated, but there is no overarching, coherent, methodological paradigm. This makes it difficult to appreciate the technical value of the contribution.
- Object recognition and grasping are well-studied research problems, but instead of leveraging some of the available, powerful methods, this system hard-wires an ad-hoc nearest-neighbor object recognizer and an ad-hoc grasp synthesizer limited to parallel-jaw grippers. Particularly the latter runs counter to a core motivation (task diversity) of this work.
- The potential benefits of user-provided metadata are not explored. This is an interesting feature that can presumably provide tangible benefit, but no detailed examples and no performance evaluation are given.
- "The main focus of this paper is the task-description problem in robotics", but no examples are elaborated where Vimex provides benefits over other ways of describing tasks, and its task-description capabilities are not evaluated at all on entire tasks.
- Vimex performance is evaluated in terms of its three core capabilities of object recognition, part localization, and affordance detection. However, all of these are well-established research problems and powerful methods exist, but no comparisons to existing methods are given.

In summary, the technical contributions of  the Vimex framework are limited, and its merits - while plausible - have not been sufficiently demonstrated.

**Questions:**

- Each of the Weaknesses implies a Question.
- I don't fully understand the "any top-K" rule. It applies in object recognition, which is usually formulated as a forced-choice, multi-class classification problem. Here however it appears to be phrased as a detection problem: The description talks about "positive" labels and mentions "detection" and "false-negative" results. However, it does not mention false positives, which I'd expect to occur abundantly under this rule, and it seems they cannot simply be fixed by taking a few images of the problematic false-positive nearest-neighbor object(s).
- Regarding the optimization issue discussed in Appendix A.4.1, it seems to me that Bayesian optimization would be a promising alternative. What would be its merits and drawbacks compared to your method?
- Why are object recognition and part localization only evaluated on the train-subsets of their respective data sets, rather than using their canonical train/test splits or by cross-validation on the entire data sets?
- Object recognition is evaluated only after SAM preprocessing because this method is considered to be external to this work. However, this framework uses SAM as one of its parts, and the original technical contribution in object recognition of this work is limited. Its main interest lies in its end-to-end performance. Thus, full-pipeline object recognition performance is also relevant.
- If I understand correctly, "particle filter" is not the best characterization of what is going on. There is no dynamic model and no resampling; it is simply a sample-based, nonparametric representation of a probability density.
- At the beginning of Sec. 6, the characterization of "visible" vs. "occluded" grasps merits improvement. For example, we can have a top-down grasp with the grasp axis parallel to both the table and the camera plane, and the camera plane perpendicular to the table surface such that both contact points are visible.
- A little suggestion: write $\exp(a)$ rather than $e^a$ to avoid tiny, hard-to-read math.

---

### Official Review · Reviewer_BqR3 · 2023-11-01

**Soundness:** 2 fair
**Presentation:** 3 good
**Contribution:** 3 good
**Rating:** 6
**Confidence:** 3

**Summary:**

The paper presents the Vimex (Visual Memex) framework, which aims to simplify the task description process for vision-based robots. Instead of engineered reward functions, or demonstrations for training neural network policies, the proposed solution aims to user-defined  scribble and metadata annotations for grasping arbitrary objects. The key idea involves using global and spatial image descriptors from DINOv2 for nearest-neighbors similarity to retrieve relevant records for object recognition, part or metadata annotation. These records are then used to define probability distributions of part locations and metadata over 3D coordinates, enabling efficient task description. The paper shows how the Vimex framework can be applied to simulated and real-world vision-based robotic grasping tasks.

**Strengths:**

1. Significance and Originality: The paper presents an important perspective on using content-based retrieval solutions to the open-world robotics manipulation problem. Vimex uses image descriptor features, extracted from models like DINOv2 and SAM trained on web-scale data, defines nearest neighbor similarity scores and applies particle filter approach to infer affordances, all of which do not require training neural networks for robotics grasping tasks.
1. Quality and Clarity: The paper is well written and the literature review is good. A detailed analysis for object recognition on COCO and part recognition on PartImageNet have been performed in the appendix. The videos for real world grasping and data for simulation experiments are provided in the supplementary material.

**Weaknesses:**

1. The paper presents results for only where grasps are antipodal on an object that is free from clutter. It seems to assume that clean images of the object have been captured and are available for annotation and grasping. Real world scenarios have occlusion, clutter and inaccurate point clouds. There seems to be a disconnect in the problem motivation and the experiment design. The claim “largely unknown environments” seems an over claim for the proposed solution, since there are object specific information and markers collected by user.

1. The 'task descriptions' in the context of this paper only refers to static grasping, and not other associated motions with the object.
The scope and current limitations should be noted in the conclusion or appendix sections.

**Questions:**

1. It seems the proposed solution would require annotation again in different orientations (in xz or yz plane, where z is along gravity) of the same object, like manipulating a soft toy for example.
1. How does this scale with the number of objects that I want the robot to interact with? The experiments suggest that one object interaction requires about ~20 MB for recognition.
1. It is unclear how does the Monte Carlo sampling based solution compares to existing approaches like contact-graspnet (whose candidate grasps can be reweighted based on user-defined annotations), particularly in terms of real-time inference speed.

---

### Official Review · Reviewer_oGXf · 2023-11-01

**Soundness:** 3 good
**Presentation:** 3 good
**Contribution:** 2 fair
**Rating:** 5
**Confidence:** 3

**Summary:**

This paper proposes VIMEX, an exemplar-based framework for open-ended "task description". The authors use pre-trained foundational vision models as object localization and representation backbones, build a memory of user-guided task-related "metadata" to define positive and negative exemplars, and then use a model-based probabilistic inference framework to connect it with affordances and robotic grasping. The approach looks promising for efficient few-shot open-ended learning in the robotic context, but there are issues regarding the scalability of the method when users define tasks, as well as regarding experimental comparison with state-of-the-art approaches for open-ended (or similarly zero-shot transferred) grasping.

**Strengths:**

- The topic of the paper is timely, given the increasing emphasis on vision in robotic applications.
- It's an interesting exemplar-based approach to scalable open-ended robot learning. Interesting adaptation of nearest-neighbor similarity to spatial-wise softmax operator and interesting casting of the affordance prediction problem to probabilistic inference.
- The paper is well-written, with an engaging Introduction that highlights the challenges of open-ended robotics, limitations of RL/IL and reconstruction-based methods and motivates the need for more abstract task descriptors. The connection to Memex is also interesting.
- It's promising to see that with 5 RGB-D images, the robot can generate robust antipodal grasps for isolated objects.
- The paper provides thorough experimental results, showcasing the potential of VIMEX in real-world scenarios.

**Weaknesses:**

- No experimental comparison with deep learning-based approaches for 6DoF grasp detection. Modern approaches can generate grasps out-of-the-box (and hence in an open-ended fashion) for unseen object instances (e.g. GraspNet).
- Similarly, object detection and part localization experiments benchmark the proposed method in COCO and PartImageNet but do not provide comparisons with any baselines.
- In general, it seems that there is a lot of effort on the user side to create their own "ontology" of object-related concepts in different task contexts, e.g. is the user expected to define the heuristic of "antipodal distances" as the object-related metadata for the grasping experiments? If not, where do those metadata come from for different contexts?
- Part localization is heavily modularized, requiring off-the-shell localization, camera pinhole, and point cloud processing to project to 3D and estimate box. All these steps introduce errors when in actual physical environments that include clutter. The authors should provide an explicit quantification of the aggregated errors from all the modules in the experiments section.
- The paper makes a case for a non deep learning-based framework for robot learning. However, throughout the paper and mainly in Sec.6, the authors make assumptions (e.g. visible vs. occluded grasp, properties of the optimization loss for inference, etc.) and integrate their method for grasping using model-based probabilistic inference. Such approaches work well in a closed environment but deep learning offers generalization. An analytical comparison and analysis are missing.
- What if you replace the singular object experiments with more natural cluttered scenes (e.g. piles), does the inference process described by the authors work robustly?

**Questions:**

- When introducing Visual Memex (Sec.4), the authors mention: "The user then decides whether the marked images should stay in D neg, moved to D pos, or removed entirely". In my understanding, this is done for all images of the negative sets that have the nearest distance to Dpos smaller than the median distance within Dpos. Given that Dneg consists of 10^6 images, you could expect a huge number of images that satisfy this constraint. Does the user have to annotate manually all the "marginal" negative examples of that size? Can the authors please explain this, if this is the case, this doesn't seem more efficient than collecting e.g. demonstration datasets?
- Authors mention in 5.1: "We, therefore, implement an “any top-K” rule: the classification label is positive if and only if there exists any positive exemplar within the top-K nearest neighbors, where K is set through cross-validation within (D pos, D neg )"
Does this have to be done for every new pair of Dpos, and Dneg? Sounds like a lot of computing if really wanted to be used in an open-ended fashion.

---

### Official Review · Reviewer_pVhr · 2023-11-06

**Soundness:** 3 good
**Presentation:** 3 good
**Contribution:** 3 good
**Rating:** 6
**Confidence:** 4

**Summary:**

The paper proposes Vimex, a general framework for task-dependent object handling and manipulation in open and unstructured scenarios. Object and part definitions are based on small collections of RGB images, and can be augmented by task-specific annotations. The framework is designed with ease-of-use and applicability in mind, as objects are simply defined through image collections depicting them and relevant object parts are annotated via scribbles on images. The framework builds on a collection of state-of-the-art (sota) foundation models for achieving individual tasks like object segmentation, part recognition etc. The framework is evaluated considering a two-finger parallel-jaw gripper grasping scenario.

**Strengths:**

The paper is well written and easy to read. The proposed framework is easy to deploy and use for the end-user, as no special training is required (e.g. neural networks, reinforcement learning, digital twins, etc.).  It addresses the challenging problem of task-dependent recognition of objects, relevant parts and affordances. It achieves this by building on descriptors based on the sota foundation models such as i) DINOv2 (Oquab et al., 2023), whose inner-product implements a semantically meaningful similarity distance  and ii) segment anything (Kirillov et al., 2023) for object segmentation. The evaluation is performed considering grasping with a two-finger parallel-jaw gripper, both for simulated and real-world experiments

**Weaknesses:**

In general, as stated in the quite comprehensive related-work section, the high-level ideas behind Vimex, and exemplar-based recognition in particular, are not new. However, new solutions are used to tackle low and middle-level vision tasks more successfully than in the past.

Another weakness is that the motivation behind the quantities and probability measures defined in Sections 5.2 and 5.3 is not entirely clear. Providing additional details behind the choice of the specific formulas would be helpful.

Importantly, the clarity of the design and execution of the experiments is quite limited. For example, how is D_neg defined for these scenarios? Is it the same used for object-recognition and part-localization evaluation? Section 4 gives Deng et al., 2009 and Li et al., 2020 as examples. Are those (or one of them) used in all cases? Additionally, a description of how is the entire pipeline operating during execution and how the individual parts interact (i.e. object, part recognition and affordance infernece) is missing. Is the pipeline continuously evaluated during execution or only from initial pose? If only the execution depends on the initial pose only, how the results depend on the initial pose?


### Minor Comments
- Figure 1 not cited in text

**Questions:**

Please consider the questions mentioned above regarding how the proposed framework is applied in the grasping experiments presented in the paper.

---

### Author Response · Authors · 2023-11-22
**Withdrawal Note**

We thank all reviewers for taking the time to provide detailed and useful feedback! They identified a number of directions related to experimental evaluation that we believe will improve our paper once incorporated. Therefore, we withdraw our paper to have sufficient leeway to properly work on them. We also noticed a number of comments that suggest a clarification here will be useful about what the main intention of our paper is and how the proposed framework sets out to achieve it. Please find these below.

## **The main point that this paper wants to convey**
The main goal of our framework is to enable a user (or a robot itself) to easily create and update an arbitrary ontology (i.e., semantic units like objects and parts) with associated metadata stored in a database (i.e., memory), to later retrieve and assign this information to the contents of an RGBD image in the form of probability distributions over 3D coordinates in the camera frame.

The emphasis is on flexibility as the metadata can constitute any information, for example about occluded geometry (e.g., antipodal distances), dynamic/material properties (e.g., mass, inertia, BSDF,  elasticity), logs of past interactions with semantic units in the ontology and deductions based on these (e.g., success/failure, graspable, hot, cold, soft, sharp), or anything else then can be measured and logged in a database.

The intent is for these 3D metadata distributions to then support many subsequent probabilistic reasoning and inference routines  for planning and control (e.g., affordance detection, tool alignment, visual servoing, 3D reconstruction of unseen surfaces, transcribing an entire intuitive physics simulation by retrieving metadata about dynamic/material properties), to generate robot behavior from RGBD images.

In short, this framework aims to provide probabilistic visual reasoning primitives that retrieve information stored in memory and project it onto the robot's current context to make it available for planning and control.

---
---
> ### *(Reviewer Comment) This paper hardwires an ad-hoc object detection method to an adhoc grasps synthesizer. There is no overarching, coherent, methodological paradigm.*

Thank you for your feedback! We believe the main reason why the current version of the paper may have created such an impression is because experimental evaluations provide only a single example case related to antipodal grasping with a two finger gripper. This is indeed not enough, and the current breadth of the experiments section should be significantly expanded to reflect all of the possibilities discussed above, to prevent creating such a misimpression in the future. We would like to emphasize that neither object detection nor grasp synthesis constitute end goals of our paper. To make this clear, we will add multiple other examples of inference (e.g., affordance detection, tool alignment, visual servoing, 3D reconstruction of unseen surfaces).

> ### *(Reviewer Comment) The paper doesn't include comparisons to state-of-the-art/deep-learning baselines for 6DOF grasp synthesis.*

All reviewers shared this criticism, and it is completely valid. We will make sure to add relevant baselines for both the current grasp inference example as well as for the other inference examples mentioned above that will be added.